# Therapeutic Strategy of Mesenchymal-Stem-Cell-Derived Extracellular Vesicles as Regenerative Medicine

**DOI:** 10.3390/ijms23126480

**Published:** 2022-06-09

**Authors:** Yasunari Matsuzaka, Ryu Yashiro

**Affiliations:** 1Division of Molecular and Medical Genetics, Center for Gene and Cell Therapy, The Institute of Medical Science, University of Tokyo, Minato-ku 108-8639, Tokyo, Japan; 2Administrative Section of Radiation Protection, National Institute of Neuroscience, National Center of Neurology and Psychiatry, Kodaira 187-8551, Tokyo, Japan; ryu.r@ncnp.go.jp; 3Department of Infectious Diseases, Kyorin University School of Medicine, 6-20-2 Shinkawa, Mitaka-shi 181-8611, Tokyo, Japan

**Keywords:** drug delivery, exosomes, extracellular vesicles, lipid nanoparticle, mesenchymal stem cell, regenerative medicine

## Abstract

Extracellular vesicles (EVs) are lipid bilayer membrane particles that play critical roles in intracellular communication through EV-encapsulated informative content, including proteins, lipids, and nucleic acids. Mesenchymal stem cells (MSCs) are pluripotent stem cells with self-renewal ability derived from bone marrow, fat, umbilical cord, menstruation blood, pulp, etc., which they use to induce tissue regeneration by their direct recruitment into injured tissues, including the heart, liver, lung, kidney, etc., or secreting factors, such as *vascular endothelial growth factor* or insulin-like growth factor. Recently, MSC-derived EVs have been shown to have regenerative effects against various diseases, partially due to the post-transcriptional regulation of target genes by miRNAs. Furthermore, EVs have garnered attention as novel drug delivery systems, because they can specially encapsulate various target molecules. In this review, we summarize the regenerative effects and molecular mechanisms of MSC-derived EVs.

## 1. Introduction

Extracellular vesicles (EVs), which are nano- to micro-sized lipid bilayer membrane particles secreted by host cells, play critical roles in novel intercellular communication mechanisms, mediating the transduction of functional molecules with physiological activity, such as microRNAs, mRNAs, proteins, and lipids (Figure 1) [1,2,3,4,5,6,7,8,9,10,11,12,13,14,15,16,17,18,19,20,21,22,23,24,25,26,27,28,29,30,31,32,33,34,35,36,37,38,39,40,41,42,43,44,45,46,47,48,49,50,51,52,53,54,55,56,57,58,59,60]. The level of EVs in humans has gained attention for the early diagnosis of various diseases using liquid biopsy owing to their distribution in various body fluids, including blood, urine, saliva, spinal fluid, and tears, as well as their stability [61,62,63,64,65,66,67,68,69,70,71,72,73,74,75,76,77,78,79,80,81,82,83,84,85,86,87,88,89,90,91,92,93,94,95,96,97,98,99,100,101,102,103,104,105,106,107]. As the amount and type of these functional molecules present within or on the surface of EVs vary depending on the disease, they could be used for disease diagnosis, prognosis, and therapeutic targets [108,109,110,111,112,113,114]. EVs are classified into exosomes, microvesicles, and apoptotic bodies, based on differences in particle size and formation mechanisms [115,116,117,118,119,120,121,122,123,124,125,126]. Since exosomes can be regarded as a natural drug delivery system (DDS) that exists in the living body, they are widely used in drug discovery technologies [127,128,129,130,131]. Furthermore, since exosomes are abundant in numerous species and play a role in the transduction of molecular information between different species, research has focused on their application in various fields and elucidation of their mechanisms in various life phenomena and health and medical care. However, exosome analysis and sample preparation techniques, which form the basis of research, are still immature. Therefore, there is a need for the development of new technologies that can facilitate a breakthrough in the research of EVs as a therapeutic strategy. In this review, we summarize the latest studies on EVs and MSCs as novel therapeutic materials.

## 2. Mesenchymal Stem Cells (MSCs) for Regeneration

MSCs are pluripotent stem cells with the ability to self-renew, regenerate, and repair deficient cells and the plasticity ability to differentiate into bone, cartilage, blood vessels, and cardiomyocytes, which are derived from the mesoderm [132,133,134,135,136,137,138]. Unlike embryonic stem cells and induced pluripotent stem cells, general stem cells, which are more abundant during early childhood than adulthood, support human growth. Stem cells, also called tissue stem cells, such as adult stem cells or somatic stem cells, are still present at maturity when apparent growth ceases and serve to replenish cells in damaged tissue throughout life [139,140]. Hematopoietic stem cells (HSCs) present in the bone marrow have been studied for more than half a century and are being actively applied clinically [141]. The establishment of a treatment method using HSCs transplantation has expanded the possibilities of transplantation using other tissue stem cells [142,143,144,145]. However, depending on the tissue, such as the brain or heart, it is difficult to separate stem cells from the living tissue and for use as treatment. In recent years, MSCs have been the focus of attention because they are relatively easy to extract from various tissues, including the bone marrow, adipose tissue, placenta, umbilical cord, synovium, and pulp [146,147,148,149,150]. Furthermore, they can also differentiate into ectoderm-derived nerve cells and glial cells that perform functions such as supporting nerve cells and endoderm-derived hepatocytes [151,152,153,154]. Thus, MSCs have garnered attention as cell sources in regenerative medicine because they grow almost indefinitely in a culture dish and perform various functions, such as wound healing, immune regulation, and nerve regeneration; additionally, the therapeutic effects of MSCs against diseases are through their paracrine action rather than differentiation into specific cells. This paracrine action—namely, immune system control, angiogenesis, anti-inflammatory effect, antioxidant action, antiapoptotic action, and tissue repair action, in which cell secretions act on neighboring cells through direct diffusion and not on endocrine cells that act on distant cells via the general circulation—involves various cytokines, including *tumor necrosis factor*-α (TNF-α), interferon-gamma (IFN-g), interleukin 6 (IL-6), interleukin 10 (IL-10), and *transforming growth factor*-β (*TGF*-β), and growth factors secreted by MSCs [155,156,157,158,159]. In particular, in MSC transplantation in cardiomyopathy, MSCs regulate the activation of *matrix metalloproteinase*s (MMPs), leading to the attenuation of cardiac remodeling [160]. In addition, MSCs produce *vascular endothelial growth factor* (VEGF), insulin-like growth factor-l (IGF-1), adrenomedullin, and hepatocyte growth factor (HGF), which stimulate myogenesis and angiogenesis in the injured myocardium [161,162,163,164,165]. Thus, MSCs improve myocardial perfusion and regeneration by differentiating into cardiomyocytes.

The minimum criteria for defining human MSCs are (1) adherence to plastics under standard culture conditions; (2) positive cell surface markers CD73, CD90, CD105, and negative CD11b or CD14, CD19 or CD79a, CD34, CD45, and HLA-DR; and (3) ability to differentiate into osteoblasts, chondrocytes, and adipocytes [166]. Since MSCs express major histocompatibility complex (MHC) class I, but not MHC class II, they are less likely to be attacked by *natural killer* (NK) cells and exhibit difficulty in developing humoral immunity. The MSCs used for clinical purposes are collected from various tissues such as bone marrow, umbilical cord, umbilical cord blood, and fat, and have important biological activities related to tissue repair, such as anti-inflammatory effect, proliferative factor secretion, and angiogenesis-promoting effects without the risk of tumorigenesis [167,168,169]. Furthermore, it is clear that the properties of MSCs differ depending on the organ from which they are collected. Adipose-derived MSCs, as well as bone-marrow-derived MSCs, have received a great deal of attention because they can be collected more easily and abundantly throughout the body, are less invasive, and have excellent organ repair and immunomodulatory abilities, compared with those of bone-marrow-derived MSCs [170,171]. Bone-marrow-derived MSCs comprise only approximately 0.01% of the cells in the bone marrow, whereas the number of adipose-derived MSCs in adipose tissue is 500 times that of MSCs in the bone marrow. Additionally, adipose-derived MSCs produce more growth factors, such as HGF and VEGF, that contribute to organ repair than those derived from bone marrow [172]. Furthermore, in addition to the ability to differentiate into fat, bone, and cartilage, similar to bone-marrow-derived MSCs, they have the ability to differentiate into the muscle, which is not derived from bone marrow. Although their cell morphology and differentiation potential are not different from those of bone-marrow-derived MSCs, they are characterized by a strong proliferative capacity, little effect of aging, and a small decrease in bone differentiation ability [173,174]. The number and growth of bone-marrow-derived MSCs decrease with age. Adipose-derived MSCs can grow sufficiently even if they are obtained from the adipose tissue of elderly individuals. General anesthesia is used to collect MSCs derived from the bone marrow, which puts a heavy burden on the patient. In contrast, when adipose-derived MSCs are collected, the burden on the patient is light because adipose tissue is close to the surface of the body. In addition, adipose-derived MSCs are characterized by a higher immunosuppressive capacity than that of bone-marrow-derived MSCs [175]. Animal studies have shown that adipose-derived MSCs can dramatically improve nephritis [176]. Moreover, MSCs accumulate at the treatment site because of the “homing phenomenon”, which includes MSCs recognizing the lesion-induced signals such as cytokines and adhesion factors. Therefore, when MSCs are injected into the blood circulation, they naturally accumulate at the desired site and exert a therapeutic effect. In particular, after intramuscular injection, MSCs deposit in the interstices of muscle fibers through the production of basic fibroblast growth factor (bFGF) and VEGF, and induce angiogenesis and support nerve cell regeneration, leading to amelioration of neuropathy [177,178,179]. These results suggest that MSCs are excellent for clinical use due to having a wide range of applications and sufficient supply, in addition to fewer safety-related and ethical issues. Allogenic MSCs are expected to have a wide range of therapeutic effects, and clinical trials are currently underway in various diseases, such as osteochondral disease, decompensated liver cirrhosis, systemic erythematosus, acute transplant-to-host disease, Crohn’s disease, myocardial infarction, cerebral infarction, and Parkinson’s disease [180,181,182,183,184,185,186,187,188,189,190,191,192]. In safety evaluation studies, mild-to-moderate abnormalities, such as fever, chills, headache, fatigue, increased anxiety, redness of administered skin, edema, weight loss, cold, and cough, were frequently observed, but no serious acute adverse events were reported even in elderly patients. Currently, regenerative medicinal products have been approved for the treatment of spinal cord injury and acute graft-versus-host disease after HSC transplantation. However, some issues still exist when using MSC for regenerative medicine.

Since MSCs, compared with autologous cells, have a higher risk of transmitting infectious factors such as viruses to patients undergoing transplantation, it is necessary to exert stringent control on the quality of the cells used as raw materials [193]. Additionally, a risk of immune rejection and low engraftment compared with that in autologous cells has been suggested [194,195,196,197,198,199]. It can be pointed out that the risk of tumorigenicity may be low because the engraftment is lower than that of autologous cells, and the immune system is easily activated. However, as the cells are amplified, the risk of accumulation of genomic mutation and chromosomal abnormalities also increases [200]. Therefore, appropriate evaluation is a critical factor because the risk of tumorigenesis varies greatly depending on the culture, proliferation period of used cells, and number of cells to be transplanted. With allogeneic cells, it is assumed that a single cell strain will be transplanted into multiple patients. Therefore, it is relatively easy to standardize and manage the timing of the processing and shipping, which leads to reduced costs. However, the risk of an intravascular embolism when administered transvascularly has been previously reported for both autologous and allogeneic cells in clinical use [201]. In addition, as an allogeneic bone-marrow-derived MSC preparation, Temcel® *HS injection* has already been approved for regenerative medicine [202]. However, the results of nonclinical studies in rats showed cell embolism in the brain, heart, lung, liver, kidney, spleen, bladder, etc., and thrombus in the lungs of some individuals [203]. Furthermore, the risk assessment in clinical studies did not rule out causality with the administered cells owing to one patient who died of gastrointestinal bleeding and one among 25 patients who exhibited a systemic rash after administration. However, no such adverse events have been confirmed in nonclinical studies to date. Another MSC product derived from allogeneic cord blood, CARTISTEM, is undergoing two clinical studies, including phase I/II and phase III, but no major adverse events have been reported yet [204].

## 3. MSC-EVs for Regeneration

MSC is a mesoderm-derived somatic stem cell that can be established from tissues such as bone marrow, fat, umbilical cord, and pulp, and has the ability to differentiate into fat, bone, and cartilage [205,206]. In addition to this differentiation capacity, MSC exerts secretory effects that induce anti-inflammatory, antifibrotic, or immunosuppressive effects. In recent years, it has been suggested that these effects are due to EVs secreted from bone marrow, fat, umbilical cord, menstruation blood, pulp, etc. [207,208,209,210]. MSC therapy is expected to be applied to various diseases including severe heart failure, but there are challenges such as individual differences and insufficient effects. Since T-cadherin, a receptor for adiponectin, is expressed in MSCs, it was revealed that adiponectin, which is secreted by adipocytes and is abundant in blood, promotes EVs production, thereby exerting a therapeutic effect on MSCs using a mouse model of heart failure (Figure 2) [211]. Thus, in MSC therapy, since the administrated MSCs produce a large amount of EVs by incorporating adiponectin into the cells via T-cadherin expressed on the membrane surface, the action of EVs on the heart improves the cardiac function of the heart failure model.

## 4. Therapy via MSC-Derived EVs as a Novel DDS System

Recent studies have shown that EVs secreted by MSCs have similar therapeutic effects on MSCs because some of the paracrine effects of MSCs are derived from exosomes, and most of the therapeutic effects of MSCs are responsible for the paracrine effects of EVs in certain diseases via miRNAs, mRNAs, and proteins as functional molecules [212,213,214,215].

As for the therapeutic effect of EVs secreted by MSCs, first, studies using animals modeled for acute renal disease showed that MSCs acted paraclinically against living epithelial cells to support tissue regeneration, in which paracrine effect plays a vital role via EV-encapsulated mRNAs involved in transcriptional regulation, proliferation, and immune regulation to induce tissue regeneration [216,217,218]. Furthermore, the in vitro signaling pathway to induce apoptosis and suppress the proliferation of renal epithelial cells was inactivated in the presence of EVs released by MSCs, leading to protection against cellular damage [219,220,221]. However, since no such effect was observed in fibroblast-derived EVs, it was assumed that the cytoprotective action is a specific property of MSC-derived EVs. Additionally, MSCs, which are induced to differentiate from ES cells or established from various fetal tissues, MSC-derived EVs have been shown to have therapeutic effects on myocardial damage in a very short time in animal models with myocardial ischemia–reperfusion disorder [222,223]. This was achieved through the delivery of proteins retaining functionality within the EVs to cardiomyocytes efficiently and rapidly, leading to reduced oxidative stress and promoted phosphorylation of the PI3K/Akt pathway; where the equivalent therapeutic effect of the EVs was indicated as an amount of 1/10 or less of the EV-depleted supernatant from the MSC culture [224,225,226,227]. In addition, the increased expression of microRNAs secreted by MSCs had therapeutic effects via the secretion of neurotrophic factors and angiogenesis-promoting factors, and the EVs ameliorate neuropathy in stroke animal models through neurite outgrowth, where administration of MSC-derived EVs intravenously had the same effects as that of cell administration [228,229]. Furthermore, in the Alzheimer’s disease (AD) model, neutral endopeptidase (NEP), which is the enzyme responsible for the rate-determining process of amyloid-beta (Aβ) in the brain of patients with this disorder and excessive accumulation of Aβ is one of the major characteristics of the pathophysiology of AD, was higher in adipose-derived MSCs than in bone-marrow-derived MSCs and contributed to the higher efficiency of Aβ degradation [230,231,232]. In addition, the NEP protein is also present in EVs from adipose-derived MSCs and exhibits enzymatic activity, leading to intracellular Aβ degradation via uptake by neural cells [233]. Moreover, in a mouse model of hypoxia-induced pulmonary hypertension, MSC-derived EVs exerted therapeutic effects by suppressing inflammation through the inhibition of the *signal transducer and activator of transcription 3* (STAT3) pathway in the lung [234]. Here, the EVs suppressed the upregulation of the hypoxia-inducible miR-17 superfamily and induced the upregulation of the growth-inhibitory miR-204, and no therapeutic effects were observed in the EV-depleted supernatant. Additionally, in a mouse model of acute lung injury, keratinocyte growth factor (KGF) mRNA, which is important for the therapeutic effect of EVs on lung disorders due to its abundance within the EVs and the paracrine effect of MSCs, is partially responsible for the healing effect on lung injury [235]. In addition, the administration of MSCs into the lungs of lipopolysaccharide-induced acute lung injury mice restored the proliferative capacity of alveolar epithelial cells and lung function through mitochondrial transmission [236]. Further, MSC-derived EVs have been reported to have opposite effects, i.e., tumor progression via tumor microenvironment remodeling and tumor suppression via regulation of immune responses and intercellular signaling. However, it has been suggested that they will be safe carriers of antitumor drugs [237,238,239].

Thus, although substantial evidence is available on the usefulness of the therapeutic effects of MSC-derived EVs, some important challenges remain. The basic molecular mechanisms of EVs, such as secretion, uptake by the receiving cell, sorting of their contents, and biogenesis, are still unclear. Various regulatory molecules in multiple molecular pathways have been identified as the mechanism of EV synthesis and secretion [240,241,242,243,244]. These pathways include the endosomal-sorting complex required for transport (ESCRT) involved in membrane vesicle formation, neutral sphingomyelinase 2/sphingomyelin phosphodiesterase 3 (nSMase2/Smpd3) that is the rate-determining enzyme for a membrane component ceramide synthesis, members of the Rab GTPase family involved in intracellular membrane vesicle transport, and heparanase that is a heparan sulfate degrading enzyme [245,246,247,248]. However, the degree of contribution of these regulatory molecules to EV biogenesis and secretion pathways varies greatly depending on the cell type. Therefore, to investigate the function of EVs in a cell, it is difficult to predict which pathways or molecules should be suppressed, even if an attempt is made to inhibit the secretion of EVs in target cells. Little is known about the mechanism of EV uptake into cells by the nonimmune cell system, except for the uptake of EVs in immune cells by phagocytosis. Another challenge is the unification of experimental techniques in EV research, especially in the isolation of EVs, including the most common techniques ultracentrifugation, separation based on molecular size via gel chromatography, and extraction reagents, which is the root of this research. It is unclear whether the EV fractions recovered by using different isolation methods show bioequivalence.

When considering the prospects for treatment strategies using MSC-derived EVs, first, a stable supply of MSC-derived EVs is necessary to identify suitable molecular pathways and appropriate management techniques to increase the yield of MSC-derived EVs while retaining their original therapeutic effects. Second, ensuring sufficient capacity for clinical application, if the therapeutic effect of MSC-derived EVs can be amplified, the therapeutic applicability will be greatly increased. A possible method to resolve this issue is to overexpress therapeutic molecules, such as mRNAs, miRNAs, or proteins, in MSC-derived EVs and produce a large number of EVs encapsulating these molecules. In fact, it has been shown that cells overexpressing certain miRNAs secrete EVs containing abundant miRNAs [249,250,251]. In addition, an approach that completely improves the therapeutic effect of MSC-derived EVs would be to produce a large number of EVs containing an excess of molecules by inducing a specific stimulus to the MSCs with the target therapeutic effects. Third, DDS techniques need to be developed to specifically deliver the MSC-derived EVs to target tissues. Since MSC-derived EVs have not been characterized completely, there is an unexpected risk from systemic administration via the intravenous route. For example, given that MSC-derived EVs can promote the repair of damaged tissues, the risk of carcinogenesis due to delivery to nontarget tissues cannot be ruled out. At present, modification of the surface of EVs seems promising for tissue-specific DDS through the expression of receptor proteins that exhibit tissue-specific expression using genetic recombination technology [252,253]. Thus, a new DDS with high specificity and delivery efficiency can be developed using MSC-derived EVs to introduce functional molecules with therapeutic effects. In contrast, since the complete biological action of EVs is not completely understood yet, the risk of side effects is a major concern. Bone-marrow-derived EVs induce dormancy of tumor cells and are involved in long-term recurrence, while EV-encapsulated miRNAs secreted by tumor cells cause intratumoral angiogenesis and promote metastasis in the brain through disruption of the blood–brain barrier [254,255,256].

## 5. Clinical Use of MSC-EVs

More than 500 clinical research on MSC is underway in the world for various target diseases, including rheumatoid arthritis, systemic lupus erythematosus, Crohn’s disease, myocardial infarction, Parkinson’s disease, spinal cord injury, osteoarthritis, and *graft-versus-host disease* (GVHD) [257,258,259,260,261,262,263,264,265,266]. MSCs other than those derived from bone marrow are also being clinically applied. Physician-led clinical trials such as amniotic-membrane- or umbilical-cord-derived MSC treatments for Crohn’s disease or acute GVHD are underway. In addition, a study of the induced pluripotent stem (iPS)-cell-derived MSCs for acute GVHD has begun. Further, bone-marrow-derived MSC Temcel® indication expanded to epidermolysis bullosa. Additionally, a clinical trial has begun in which pulp-derived MSCs are administrated for acute cerebral infarction. Since MSCs are slightly different in nature depending on the organization from which they are sourced, it is important to select a cell source suitable for specific diseases and therapeutic effects. Moreover, it has been clarified that the secreted EV plays an important role when MSC exerts various actions. Additionally, it has also been reported that administration of EVs secreted by cultured bone-marrow-derived MSC to patients with refractory GVHD resulted in improvement in symptoms. Furthermore, clinical trials of MSC-derived exosomes are currently underway for diabetes mellitus type 1, cerebrovascular disorders, coronavirus, Alzheimer's disease, and osteoarthritis (Table 1).

## 6. Conclusions

MSC-derived EVs treated with the cytokines upregulate the expression of the immunomodulatory molecules, including miRNAs and proteins, involved in the immunoregulatory pathways. This plays an important role in tissue repair of chronic damage through the concentration of active ingredients in the contents and efficient migration of the macrophages incorporated in the MSC-derived EVs to the damage site due to the removal of dead cells and improvement of fibrosis. These therapeutic effects are equal to or higher than those of the MSCs themselves. These reports suggest that the administration of MSC-derived EVs is a useful novel cell-free therapeutic strategy.

## Figures and Tables

**Figure 1 ijms-23-06480-f001:**
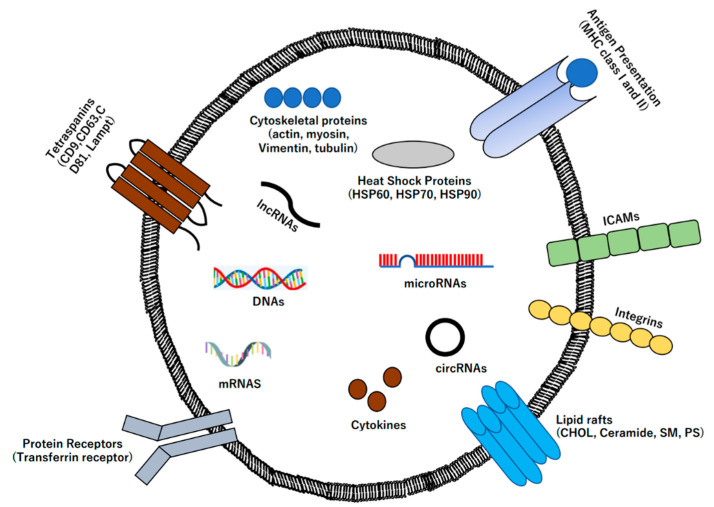
Extracellular vesicle structure. Exosome consists of lipid bilayer membrane, including proteins such as cytoskeletal proteins (actin, myosin, vimentin, tubulin, etc.), heat shock proteins (HSP60, HSP70, HSP90, etc.), tetraspanins (CD9, CD63, CD81, etc.), cytokines (IL-1β, TNF-α, IL-6, etc.), Lamp, nucleic acids such as microRNAs, circRNAs, IncRNAs, lipid rafts such as cholesterol, ceramide, sphingomyelin, phosphatidylserine, and protein receptor such as transferrin receptor.

**Figure 2 ijms-23-06480-f002:**
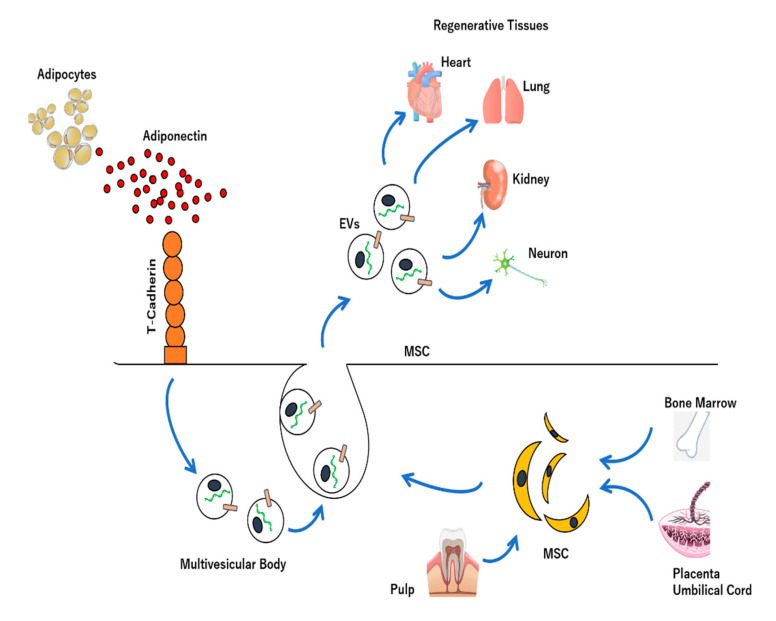
Regeneration effects of MSC-derived extracellular vesicles (MSC-EVs). MSC-EVs have different sources, including adipocytes, bone marrow, umbilical cord, pulp, etc. These MSC-EVs represent regenerative effects for heart, lung, kidney, neuron, etc. Further, MSC promotes secretion of EVs via interaction between T-cadherin receptor on the MSC and adiponectin derived from adipocytes, leading to regeneration effects of injured tissues.

**Table 1 ijms-23-06480-t001:** Clinical trials of MSC-derived Exosomes.

#	NCT Number	Condition or Disease	Phase	Sponsor	Brief Summary
1	NCT02138331	Diabetes Mellitus Type 1	Phase 2, Phase 3	General Committee of Teaching Hospitals and Institutes, Egypt	Effect of Microvesicles and Exosomes Therapy on β-cell Mass in Type I Diabetes Mellitus
2	NCT03384433	Cerebrovascular Disorders	Phase 1, Phase 2	Isfahan University of Medical Sciences, Iran	Allogenic Mesenchymal Stem Cell Derived Exosome in Patients With Acute Ischemic Stroke
3	NCT03437759	Macular Holes	Early Phase 1	Tianjin Medical University, China	To assess the safety and efficacy of mesenchymal stem cells (MSCs) and MSC-derived exosomes (MSC-Exos) for promoting healing of large and refractory macular holes (MHs).
4	NCT03608631	Metastatic Pancreatic Adenocarcinoma	Phase 1	M.D. Anderson Cancer Center, US	iExosomes in Treating Participants With Metastatic Pancreas Cancer With KrasG12D Mutation
5	NCT04173650	Dystrophic Epidermolysis Bullosa	Phase 1, Phase 2	Aegle Therapeutics, US	MSC EVs in Dystrophic Epidermolysis Bullosa
6	NCT04276987	Coronavirus	Phase 1	Ruijin Hospital, China	A Pilot Clinical Study on Inhalation of Mesenchymal Stem Cells Exosomes Treating Severe Novel Coronavirus Pneumonia
7	NCT04313647	Healthy	Phase 1	Ruijin Hospital, China	A Tolerance Clinical Study on Aerosol Inhalation of Mesenchymal Stem Cells Exosomes In Healthy Volunteers
8	NCT04388982	Alzheimer Disease	Phase 1, Phase 2	Ruijin Hospital, China	Safety and the Efficacy Evaluation of Allogenic Adipose MSC-Exos in Patients With Alzheimer's Disease
9	NCT04491240	SARS-CoV-2 PNEUMONIA	Phase 2	State-Financed Health Facility, Russia	Evaluation of Safety and Efficiency of Method of Exosome Inhalation in SARS-CoV-2 Associated Pneumonia
10	NCT04602442	SARS-CoV-2 PNEUMONIA	Phase 2	State-Financed Health Facility, Russia	Safety and Efficiency of Method of Exosome Inhalation in COVID-19 Associated Pneumonia
11	NCT04747574	SARS-CoV-2	Phase 1	Tel-Aviv Sourasky Medical Center, Israel	Evaluation of the Safety of CD24-Exosomes in Patients With COVID-19 Infection
12	NCT05060107	Osteoarthritis, Knee	Phase 1	Universidad de los Andes, Chile	Intra-articular Injection of MSC-derived Exosomes in Knee Osteoarthritis
13	NCT05216562	SARS-CoV2 Infection	Phase 2	Dermama Bioteknologi Laboratorium, Indonesia	Efficacy and Safety of EXOSOME-MSC Therapy to Reduce Hyper-inflammation In Moderate COVID-19 Patients
14	NCT05261360	Knee; Injury, Meniscus	Phase 2	Eskisehir Osmangazi University, Turkey	Clinical Efficacy of Exosome in Degenerative Meniscal Injury
15	NCT05402748	Fistula Perianal	Phase 1, Phase 2	Tehran University of Medical Sciences, Iran	Safety and Efficacy of Injection of Human Placenta Mesenchymal Stem Cells Derived Exosomes for Treatment of Complex Anal Fistula

Searched by ClinicalTrials.gov (https://clinicaltrials.gov/ct2/home, accessed on 1 June 2022).

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
