# Peer review of "Therapeutic Strategy of Mesenchymal-Stem-Cell-Derived Extracellular Vesicles as Regenerative Medicine"

_ijms, 2022, doi:10.3390/ijms23126480_

Round 1

Reviewer 1 Report

Matsuzaka et al reviewed the therapeutic strategy of mesenchymal stem cell-derived EVs in the regenerative medicine field. The review was prepared well, and I recommend the manuscript for possible publication.

Author Response

Thank you for your positive comments.

Reviewer 2 Report

In the present study, Matsuzaka et al. summarized MSC-EV for treating diseases for application in regenerative medicine. Although it is interesting, there are some comments for the authors.

  1. After the section of “2. Mesenchymal Stem Cell (MSC) for regeneration”, it is necessary to add a part for MSC-EV for regeneration, and difference sources of MSC-EV should be mentioned and if you can compared with them, it will be better.
  2. It is necessary for present the role of MSC-EV in various diseases, and the underlying mechanism can be summarized by a figure.
  3. “4. Adeno-associated virus (AAV) vector in MSC-mediated immunomodulation”, this part has no relation to MSC-EV in regenerative medicine, please re-write this part.
  4. “MSC-derived EVs treated with the cytokines IFN-γ and TNF-α upregulate the expression of the immunomodulatory molecules”, whether authors have strong evidence, and it should have more cytokines to exert the immunomodulatory function.
  5. The content is not enough, and the depth of MSC-EV for regeneration can be improved.
  6. What is the novel point in your review? All information is well-known. It is better to provide some novel outlook in MSC-EV for regenerative medicine.

Reviewer 3 Report

This review summarizes the latest studies on EVs and MSCs as novel therapeutic materials

The manuscript is well  written and the studies are appropriately described. 

However, the following points should be addressed.

Major comments:

  1. The figure 1 describes the Exosome structure but in the text the authors always describe the EVs

  2. In paragraph 3 the authors didn’t cite the works about antitumor effects of extravescicoles  isolated from mesenchymal stem cells loaded with chemotherapies. They should add them.

  3. In this review a paragraph specific for the characterization about EV for clinical use should be added

Minor comments

  1. the references [180-184] are not enough do declare that “Allogenic MSCs are expected to have 134 a wide range of therapeutic effects, and clinical trials are currently underway in various diseases, such as osteochondral disease, decompensated liver cirrhosis, systemic erythematosus, acute transplant-to-host disease, Crohn’s disease, myo- cardial infarction, cerebral infarction, and Parkinson’s disease”

  2. also the reference 186 is not enough and the authors should add also other important works which described the  risk of immune rejection and low engraftment

Reviewer 4 Report

I accept the manuscript in present form

Author Response

Thank you for your positive comments.

Round 2

Reviewer 2 Report

In the present study, although they made some revision, there are some comments for the authors. 1. Figure 2 should be re-arranged. Figure 2 only mark heart, and the caption has not mentioned it. More tissues rather than only heart are shown in Figure 2. 2. “3. MSC-EV for regeneration”, I suggest authors present the different sources of MSC-EV, including bone marrow, fat, umbilical cord, menstruation blood, and pulp etc. These references are recommended for you (PMID: 34479611, 34344458, 34449901, 35276207). 3. The challenge of MSC-EV application can be added after the section 5. 4. Abstract should be further improved.

Reviewer 3 Report

The authors have considerated all suggestions.

Now the manuscript can be published -

(In the figure 2 ther is typing  error )

Author Response

Thank you for your positive comment.